# Poverty-Aware Programs in Social Service Departments in Israel: A Rapid Evidence Review of Outcomes for Service Users and Social Work Practice

**DOI:** 10.3390/ijerph20010889

**Published:** 2023-01-03

**Authors:** Shachar Timor-Shlevin, Yuval Saar-Heiman, Michal Krumer-Nevo

**Affiliations:** 1The Louis and Gabi Weisfeld School of Social Work, Bar Ilan University, Ramat Gan 5290002, Israel; 2The Spitzer Department of Social Work, Ben-Gurion University of the Negev, Beer-Sheva 8410501, Israel

**Keywords:** critical practice, evaluation, Poverty-Aware Paradigm, review, social services departments

## Abstract

Critical perspectives and practices are fundamental to social work, yet there are only scarce examples of direct critical practice in public social services, and even fewer empirical evaluations of their outcomes for service users and social workers. This article presents a rapid evidence review of 25 evaluation studies of five programs that operate in the social services departments in Israel according to the principles of the Poverty-Aware Paradigm (PAP). The PAP is a critical paradigm for direct social work practice with people living in poverty that was implemented in the welfare services by the Ministry of Welfare, targeting over 14,000 service users. The evaluation studies we reviewed encompass an overall quantitative sample of 4612 service users and 1363 professionals, and a qualitative sample of 420 service users and 424 professionals. The findings present: (1) the program’s outcomes for service users in terms of relationship with social workers, financial circumstances, family relations, and children’s safety; and (2) the program’s impact on social workers’ attitudes and practices. Finally, we discuss the lessons learned regarding social workers’ role in combatting poverty, the construction of success in interventions with people in poverty, and the article’s limitations.

## 1. Introduction

While various proposals for radical and critical practice have been advanced throughout the history of social work, their implementation has usually been limited to small-scale programs that take place outside of public social services frameworks [1,2]. Moreover, these programs have not always been evaluated, so their impact on service users and the professionals involved has remained unclear. This article is based on the evaluation studies of a current paradigm of critical social work: the Poverty-Aware Paradigm (PAP) [3], which has been widely implemented in the public social services departments in Israel since 2015.

The PAP is a unique case in the field of critical social work. First, it bridges the gap between macro- and micro-level practice in social work and offers a detailed framework for direct social work practice that adds to the established strategies of macro-level practice [4,5]. Second, the PAP’s focus on poverty as a central axis of analysis and intervention contributes to the body of critical social work knowledge, which has mainly addressed issues of gender and race. Third, since 2015, the PAP has been implemented in nine programs that operate in half of the public social services departments (SSD) in Israel. This process, which was initiated by the Ministry of Welfare and Social Affairs [6], has included major changes in the training and supervision of social workers as well as in the organizational setting [7]. The PAP program aims to have an impact on both service users and the social work profession.

This article presents a rapid evidence review of current PAP research and evolving evaluation studies [8]. It begins with a brief introduction of the PAP’s conceptual framework and methodology. Next, we present the evaluation of the PAP as it emerged from 25 studies. This section presents the direct impact of the PAP programs on service users, social workers, and social work practice. Finally, we discuss the lessons learned from our review regarding social workers’ role in combatting poverty, the construction of success in intervention with people in poverty, and the article’s limitations.

### 1.1. Poverty-Aware Social Work

As a paradigm, the PAP contains comprehensive theoretical, ethical, and practical principles. Since these have been detailed elsewhere [3,6,9], we present them here only briefly. Following the critical school of poverty studies [10,11,12], the PAP conceptualizes poverty as a violation of human rights in the realms of both structural opportunities (e.g., opportunities for housing, employment, education, and health) and interpersonal relations (e.g., opportunities to receive recognition as equal human beings and be respected and valued). The PAP emphasizes the ongoing everyday resistance of people to their poverty. Everyday actions of resistance are seldom recognized as such since they contradict the hegemonic perception that blames people for their circumstances [13]. For social workers to be relevant to service users, they should *stand by* them in their efforts to resist poverty [3] (p. 1803). To do this, social workers must deconstruct power relations and not encourage service users to adapt to unjust contexts. In practical terms, the PAP is based on a combination of intensive relational-based and rights-based practices. This combination allows social workers to recognize and respond to the emotional and material needs of service users [9,14,15].

### 1.2. The Implementation of the PAP

The PAP was developed by the third author in the framework of a small-scale fieldwork training program for social work students that was established in 2010 in Ben-Gurion University of the Negev (this program will be referred to as P1). From 2015 to 2017, the Ministry of Welfare began to implement the PAP in four special programs in social services departments. All these programs aimed to provide families in poverty with intensive, holistic care for two years. The first was MAPA, a pilot program for families regarded as out of reach (P2) that involved 50 social workers who worked with 150 families according to PAP principles in addition to their regular caseload. In 2016, a large-scale program called Families First (P3) was established and involved some 770 social workers and 14,000 families. By 2017, in light of promising data from these pilot programs, the Ministry of Welfare decided to adopt the PAP as the leading professional paradigm for working with people living in poverty and initiated two additional PAP programs. These programs were unique because they targeted families in poverty whose children were in the child protection system (P4, P5) [16].

To implement the PAP’s theoretical guidelines and practice, all of these PAP programs include, with some variations, five organizational components [3]. The social workers undergo special *training* and receive *ongoing PAP supervision*. The training focuses on developing a critical-structural standpoint in social workers regarding poverty as well as the critical reflexivity required to overcome othering and recognize service users’ subjectivities. The ongoing PAP supervision focuses on supporting the fragile and often stormy relationships between social workers and service users and provides social workers with a supportive space in which to hold the immense pain of injustice and use critical reflexivity [6].

In addition to the special training and supervision they received, the PAP social workers had *limited caseloads* to allow them the space and time to develop close and intensive “standing by” relationships. In addition, each family that participated in the programs received a *personal budget* that was managed by their social worker to help with payments for various needs such as food, housing, repayment of debts, basic equipment, or professional assistance. This budget was flexible and enabled the social workers to materially operationalize the concept of standing by service users. Finally, a new role—“rights social workers”—was developed. The new rights social workers were trained to navigate the complex terrain of rights and engage in policy practice, and they worked alongside the frontline social workers and assisted them in the actualization of service users’ rights [6].

Currently, nine PAP programs, some thoroughly integrated into the system’s operation and some in different pilot phases, are operating in the SSDs. In addition, the PAP is currently being integrated into other social services in the field of mental health, the legal system, and probation services [6]. Our study will be limited to the five programs that were initiated up to 2017 because the newer ones are still undergoing evaluation. See Table 1 for an overview of the programs.

Our research questions were: (1) *What are the outcomes of the five PAP programs for service users?* and (2) *What kinds of changes in social workers’ attitudes and practices have occurred following the PAP’s operation?*

## 2. Methodology

We conducted a rapid evidence review, which applies a systematic approach to evidence identification and synthesis with a more limited scope than a systematic review [8]. We chose this method for two reasons. First, the implementation of the PAP is relatively recent and, more importantly, still evolving. Thus, there is a need for a rapid process to capture current evidence. Second, since the paradigm is implemented in five distinct programs, the research on it is conducted in a very specific context and is in its early phases. Accordingly, the broad search and the methods used to undertake a systematic review are not required for scoping the evidence.

### 2.1. Inclusion/Exclusion Criteria

The following criteria were developed to bring together all the viable evidence on the operation of the PAP programs: (a) empirical (quantitative, qualitative, mixed methods) studies, (b) publication in peer-reviewed or grey literature outlets, (c) publication in Hebrew or English, (d) exploration and examination of interventions that explicitly embraced the PAP, and (e) the involvement of distinct features of the PAP in direct practice and at the policy, organizational, and managerial levels. Thus, studies that discussed the PAP without examining its application in practice were excluded, as were studies that referred to poverty-aware aspects as one of many components of an intervention.

### 2.2. Search Strategy

We began by running preliminary searches on academic (e.g., Web of Science) and nonacademic (e.g., Google Scholar) databases. The search terms we used were poverty AND aware AND (“social work” OR practice OR policy) AND implementation. An initial screening of the results revealed that all the studies that met our inclusion criteria referenced either Krumer-Nevo (2016) [3] or Krumer-Nevo (2020a) [9]. This is because the inclusion criteria required an explicit embracing of the PAP. Therefore, we decided that the search would focus on the 172 resources mentioned in Google Scholar as citing Krumer-Nevo, 2016 and 2020a.

Since the implementation of the PAP takes place predominantly in the Israeli context, a search of peer-reviewed and grey publications in Hebrew was also conducted. Two strategies were employed to search for research in Hebrew. First, we searched the Index to Hebrew Periodicals, which provided 18 results. Second, we contacted the senior manager at the Ministry of Welfare and Social Affairs who oversaw the implementation of the PAP in all the programs. All studies undertaken on programs implemented by the Ministry require her approval, so she is a reliable source and referred us to six studies that were commissioned by the Ministry to evaluate the programs and not published in academic journals. Thus, the initial sample contained 196 publications.

### 2.3. Screening, Study Selection, and Data Extraction

After removing 46 duplicates (there was an overlap between Hebrew and English publications and between publications that cited [3,9]), we conducted a title and abstract screening. In this screening we excluded 111 publications that were conceptual or did not focus on PAP programs in Israel. Next, we conducted a full-text screening of 39 publications and, following a discussion, agreed to exclude 14 of them that focused on various aspects of PAP programs, but not on their outcomes. The final sample consisted of 25 studies. All three authors defined the inclusion and exclusion criteria. Yuval Saar-Heiman led the first screening and exclusion of duplicates. The abstract screening was conducted by all three authors, as well as the in-depth reading of the full-text publications. We consulted two independent researchers regarding the screening process. See Figure 1 for the screening process.

### 2.4. Data Synthesis

After extracting key data (e.g., research site, methods, sample characteristics and size, and summary of the findings) from each publication, we employed a narrative approach to synthesis. Narrative synthesis relies primarily on text to summarize and explain the findings of multiple studies [17]. It involves summarizing individual studies, grouping them according to relevant characteristics, and identifying commonalities and differences within and between groups.

After discussing the initial findings from the data extraction, the authors decided to synthesize the evidence based on two categories: evaluation of the PAP intervention outcomes for service users and evaluation of the PAP’s impact on social workers’ attitudes and practices.

### 2.5. Overview of the Sample

The final sample used for this article consisted of 25 publications that stemmed from 16 research projects/datasets. All the studies were undertaken in Israel and focused on the implementation of the PAP at different sites, i.e., P1 (n = 2), P2 (n = 11), P3 (n = 17), P4 (n = 4), P5 (n = 2), and training programs (n = 1). Eight studies [18,19,20,21,22,23,24,25] were initiated by the Israel Ministry of Welfare, which published research reports on them in Hebrew. One more research report was initiated and published by the Taub Center for Social Policy Studies in Israel [26]. The rest of the publications in the sample (n = 16) appeared in peer-reviewed outlets.

Regarding the methods used to investigate the operation of the PAP, the majority of studies (n = 14) utilized qualitative methods, eight studies used mixed methods, and three were quantitative. The main qualitative method used was semi-structured interviews, e.g., [27,28], while other methods, such as focus groups, e.g., [18,27] and ethnographic observations [29] were also evident. Overall, approximately 424 social workers and 420 service users took part in studies that employed qualitative methods.

The quantitative studies included one randomized controlled trial [23] (with 224 service users and 100 social workers), six studies based on questionnaires (approximately 4612 questionnaires were filled in by service users and 1363 by social work practitioners and managers), and three administrative data analyses, e.g., [24,26] that included approximately 5700 service users. Importantly, all the quantitative designs involved data collection at two or more points in time to explore changes and outcomes over time. Since the COVID-19 pandemic broke out soon after the first wave of families had completed the two-year programs, it was difficult to obtain an accurate picture of the families’ post-program status.

While all the studies address the implementation and operation of the PAP, their focuses varied. Ten studies were either identified by their authors as evaluation studies or focused on intervention outcomes, eight focused on one specific feature of PAP practice (e.g., material assistance, active realization of rights), four addressed the organizational and policy context of implementing the PAP, one focused on training, and one addressed the PAP in Arab-Palestinian society. See Table 2 for an overview of the sample.

## 3. Findings

The review sample pointed to the PAP programs’ outcomes for service users and the impact of the PAP programs on social workers’ attitudes and practices.

### 3.1. Evaluation of Outcomes for Service Users

The studies that evaluated the outcomes of the PAP programs for service users addressed outcomes in four main realms: (1) service users’ relationships with social workers and their willingness to receive professional assistance, (2) service users’ financial circumstances, (3) family relationships, and (4) children’s safety and wellbeing.

#### 3.1.1. Service Users’ Relationship with Social Workers and Their Willingness to Receive Professional Assistance

All the studies demonstrated strong indications of improvement in the relationships between service users and social workers. This was true regarding all the programs both in studies that focused on service users’ perspectives and those that focused on social workers’ perspectives. Examined using both quantitative [23] and qualitative measures [18], service users were highly satisfied with the treatment they received in the PAP programs, including the largest one, Families First (P3), which targeted the general population of families in poverty, and those that targeted more vulnerable families. Even the most hard-to-reach service users—parents whose children were at risk of out-of-home placement (P4, P5)—reported very positive experiences with the social workers and noted that they were different from all their previous social workers. One service user said the following, “You know that there is someone with you, that you are not alone. You are not ignored, which is the hardest thing. She helps me a lot and it really warms my heart” [24] (p. 25). The attitudes of service users in P2 who had refused contact with social workers before entering the program changed and they perceived the social workers in the program as non-judgmental, supportive, and caring in ways that restored their trust, as this quote demonstrates: “The social worker made me regain my trust. Because I [had] lost trust in the welfare department and in social workers” [18] (p. 8).

A common experience reported in many of the studies was service users’ feeling that the social workers were not only interested in their children’s situation but also in the well-being of their parents. One father in P4 said, “I had never experienced anything like that. Never. I will never forget how the social worker took me, really hand in hand, she picked me up with her old car, and took me to the National Insurance Institute, saying, ‘Don’t worry, I’ll take care of you’” [24] (p. 25). Moreover, the evaluation of P4, which used a randomized controlled trial, found that families in the program began to receive more help from other professionals than families in the control group, with a statistically significant difference [23].

Although this outcome was consistent in all the programs and studies, it may be interpreted as a basic outcome of the extra time and resources invested in the families in the framework of the special programs. Importantly, higher satisfaction among social services users was found to be correlated not only with participation in the special PAP programs but also with the kind of training and supervision the social workers received. An examination of 235 service users treated by 50 social workers in 11 welfare departments revealed higher rates of satisfaction among service users treated by social workers who underwent PAP training and supervision outside the PAP programs than in service users treated by social workers who did not receive PAP training and supervision [32]. Since the resources invested in these two groups were the same limited ones that the welfare departments provided, this finding served as an important indication of the significance of the social workers’ approach that goes beyond the resources they provided. In other words, this research indicates that the PAP professional position of standing by, which is built through training and supervision, is significant in creating a beneficial working alliance with people living in poverty.

An examination of social workers’ perspectives reveals a mirror image of this picture. For example, social workers in P2 shared the following: “Although I have known these families for years, this was the first time I really heard their story” [18] (p. 7) and “The contribution of the program is our ability to pause before judging service users and our ability to understand their life contexts” [18] (p. 7). In addition, social workers reported that they had loosened their professional boundaries and noted that the PAP practice enabled them to develop ways to work with families previously perceived as uncooperative or as constituting a potential risk to their children [18].

Despite the improvement in the relationships between social services users and social workers, the studies indicated that the two-year timeframe of the special PAP programs is not sufficiently long. For example, one evaluation study of P3 found that 75% of the families felt high confidence in the SSD’s assistance towards the end of the first wave of the program. However, two years later, 45% of the families that had completed the program reported they were not receiving the help they needed from the SSD, and only 18% of these families were in contact with the program staff [21]. One service user said: “After the program, everything stopped, I was thrown into deep water… and I felt like I was literally drowning. A few months later, I was depressed because there was no one to accompany me” [21] (p. 31). This pre-determined endpoint was the result of budgetary and political considerations and did not emerge from the families’ needs or developmental processes. All the program evaluations indicated that this timeframe did not suit the families (see, e.g., [24] (p. 25)).

#### 3.1.2. Service Users’ Financial Circumstances

There were significant differences between the programs in terms of defining improvement in families’ financial circumstances as a goal. While P3 did define it as a major goal, other programs that were aimed at families of children at risk defined the improvement of children’s well-being and the reduction of risk levels as major goals. Nonetheless, increased income was found in all the programs, even those that targeted families with children at risk, albeit to varying degrees. In the first evaluation of P3, Dank [19] (n = 346 families) found an overall improvement of between 52% and 65% in household income, a 20% increase in household employment, and a 10% increase in salaries for men and 17% for women for the same average of working hours. These findings are explained as indicating an increase in net salary per hour. Other employment indicators also increased. For example, there was a 56% improvement in participants’ perspectives regarding job suitability and a 55% improvement in job satisfaction. In the second evaluation study of P3 [20] (n = 2029), an increase of approximately NIS 950 (equivalent to approximately USD 275), i.e., 16%, was found in the household income of families after two years in the program, compared to families in their first six months in the program.

Another study that compared participants in P3 at two time points to those in a synthetic control group with similar characteristics found significant improvement in family income (11.4%) and employment status (increase in percentage of employed participants, working hours, and percentage of participants with vocational training) and a decrease in the percentage of participants with debt in the P3 group, compared to no significant changes in these parameters in the control group with standard care [28]. In P2, social workers reported that while participating in the program, 28% of the families improved in terms of employment and 23% improved their financial situations [18]. The picture was similar in P4, where there was a significant improvement in the financial circumstances of 43% of the children in the PAP program group in comparison to 17% of the children in the control group. The families’ budget management also improved (41% in the PAP group, 28% in the control group). In addition, there was a significant difference between the groups in terms of employment, with 20% of the families in the PAP program in which at least one of the parents started to work, in comparison to only 3% in the control group [23]. In terms of the sustainability of these outcomes, an evaluation of 290 families after completing P3 found a continuous increase in family income compared to families at the middle of the program (12% increase), and high levels of employment two years after completion of the program, mainly for women; 84% of the women who worked were still working [21].

In regard to programs targeting families with children at risk, for example, an evaluation of P4 (n = 46) detailed a significant increase in families’ income and financial situation at the end of the program compared to their situation at the beginning of the program [24]. In addition, the evaluation of P2 found that families had entered a debt settlement process (12% of the families in the first round of the program and 58% in the second round) [18].

These improvements, however, cannot serve as an indication of escaping poverty. The second evaluation of P3 among 2029 participants [20] found that the financial circumstances of families that completed the program remained fragile. For example, families’ debt rates were relatively high at the end of the program (an average of approximately USD 20,500), although nearly half were in the process of settling their debts. Furthermore, the families reported that they had to give up significant basic needs due to a lack of resources (42% gave up food, 49.4% gave up home repairs), and 11.5% reported a high chance that they would have to vacate their homes.

#### 3.1.3. Family Relationships

Indicators of family relationships were examined mainly in the programs that targeted families with at-risk children, and a significant improvement was found in various indicators of family relationships. In P4, there was a significant improvement of 45% in the relationship between the parents in the PAP program group and an improvement of only 25% in the control group. The PAP group reported a 40% decrease in family violence compared to a 24% decrease in the control group, and a 41% increase in families’ internal support compared to a 22% increase in the control group [23] (p. 34). Another study of P4 showed there was a significant decrease between T1 (the beginning of the program) and T2 (after two years in the program) in terms of parents’ alcohol abuse, from 50% of the families having at least one parent with an alcohol problem at T1 to 39% of the families at T2. There was also an improvement in parameters of parental functioning (from 47% of the parents protecting their children from risk at T1 to 70% at T2) [24]. In P5, a significant decrease in parent–child conflicts was found from the beginning of the program and after one and a half years in the program. Similarly, a significant increase was found in families’ ability to cope with stress [25].

#### 3.1.4. Children’s Safety and Well-Being

Children’s well-being was examined in the PAP programs for families with children in the child protection system (P4, P5), and significant improvements in various indicators were found. In P4 there was a significant difference between the PAP program families and the control group families in terms of improvement in protection from risk situations (36% in comparison to 21%) and the level of parents’ daily care and protection of their children (35% in comparison to 28%), decrease in violent behavior among children (34% in comparison to 24%), and relationships with friends (34% in comparison to 20%) and adults (31% in comparison to 16%). In addition, there was a significant difference in various dimensions of school performance, such as verbal ability, language use, and fine motor skills [23]. Significant improvements were also found for young children aged 0 to 7 whose families participated in P4, with improved outcomes in almost all measures of functioning, for example, an 11% increase in socio-emotional functioning, a 15% increase in school functioning, and a 14% increase in age-appropriate school performance from the beginning and end of the program. Furthermore, there was an increase in the number of families in which there was no suspicion of violence during the program [24].

The evaluation study of P5 examined parameters of child neglect at four time points, with T1 representing the family’s entrance to the program, and the next three Ts every six months up to a year and a half into the program. A similar pattern was evident in all parameters of neglect, indicating an increase in neglect reports between T1 and T2 (after six months in the program) and a decrease in neglect reports between T2 and T4 that eventually reached a statistically significant lower rate of neglect than at the beginning of the program [25] (p. 68). The researchers explain the increase in neglect rates between T1 and T2 as reflecting higher levels of connection with the welfare system and visibility of the families within it, and the decrease between T2 and T4 as reflecting the impact of the intervention itself [25]. Social workers’ perceptions of these improvements were revealed using qualitative methods, as this quote demonstrates: “At first, they [the kids] were far away from her. She was here and they were there. Disrespected her. Suddenly they give her a hand, listen to her … as if she’s back to being their mother” [25] (p. 152).

An examination of the children’s circumstances six months after the families completed P4 indicated that among most of the children, these outcomes remained constant. The situations of the children in the PAP group families were better than those of the control group in at least half of the measures. For example, there were significant differences between the groups in school attendance (98% for the PAP group, 90% for the control group) and reports of emergency events such as abuse or neglect, children’s self-abusive behavior, or abusive behavior toward others (12% of the children in the PAP group, 23% of children in the control group) [23].

Although the aim of the programs for families in the child protection system was to prevent out-of-home placements of children, the evaluation of P4 indicates that there was no significant difference between the PAP group and the control group in this regard. However, the decisions regarding children were agreed upon by the parents and social workers together, due to the change in their relationships and attitudes toward each other [24] (pp. 51–52). As one mother stated, “I did it only for [the child]. Because I knew that I couldn’t make it with two children. So this is my truth … because if I had lied and made it seem better, maybe she would have been with me and everything would have collapsed, and this is not the goal.” Another mother said the following: “This program gives you everything, everything, and if you don’t succeed, you are really not capable of being good for your child. You tried everything and you can feel you did the right thing” [24] (pp. 51–52).

### 3.2. Evaluation of the PAP’s Impact on Social Workers’ Attitudes and Practices

Implementing critical principles of practice requires professionals to critically analyze social problems and develop critical reflectivity regarding power relations in their practice [39]. This section presents the findings regarding changes in attitudes among professionals and changes in the actual practice they conduct. In terms of practice, we discuss the adoption of relationship-based and rights-based practices according to the PAP model [9].

#### 3.2.1. Social Workers’ Attitudes toward Poverty and People in Poverty

All the studies that examined social workers’ attitudes found that they developed a degree of critical analysis during the PAP training and ongoing supervision, e.g., [31,33]. A quantitative study that investigated the attitudes of 127 social workers before and after PAP training courses found significantly higher scores in the perception of poverty as structural and in acknowledging the relational and symbolic aspects of living in poverty at T2 [31]. In P2, 33 social workers were interviewed before, during, and after the three-year pilot phase of the program, showing that they developed structural and holistic interpretations of service users’ stories that connected closely to the relational and symbolic meanings of poverty: “[Previously] I thought that the children should leave the house … I learned to see them first of all as one family … and as partners” [18] (p. 10).

A comparison of studies that evaluated the first steps of the PAP’s implementation with more recent studies revealed a change in how social workers and social services have integrated the PAP perspective into their work. In the initial phase of the PAP implementation, there was notable tension between social workers who worked in the framework of the PAP programs and their colleagues from the same teams who did not and who held conservative views of service users as “welfare cheats” [40] (p. 235). In a study that examined social workers’ perspectives from 2015 to 2017 in P2 and P3, a social worker said, “We always feel these clashes [between the social workers in the team]… For instance, with my manager, it feels as if it’s really hard for her to give material support … She would ask, ‘Why, how is this happening again, why doesn’t he work?’ … as if they constantly suspect service users are cheating them” [34] (p. 959). Following these initial evaluations, the Welfare Ministry initiated PAP training for wider circles of social workers to cover not only those who worked directly in the PAP programs but also their colleagues in the same SSDs [6]. Indeed, studies conducted four to five years after the initiation of the PAP programs found that the PAP’s perspective was much more accepted and rooted in the field than previously. A study that examined the development of the PAP in P2 and P3 using a case study design in 2020–2021 found that poverty-aware perspectives had been adopted as the basic values of the SSD examined, affecting the physical setting of the department, the operation of decision-making forums, and the overall poverty discourse at the municipal level [29].

#### 3.2.2. Social Workers’ Implementation of PAP Practice

Poverty-aware practice is constructed as the combination of relationship-based practice and rights-based practices of material assistance and the active realization of rights [9].

Relationship-based practice: All the studies pointed to the development of intensive intimate helping relationships, e.g., [4,24,37,38]. Various practices were connected to achieving close helping relationships. In P2, social workers reported being more available to families, meeting with them more frequently (in 77% of the families in comparison to 23% of the families before the program began), and conducting more home visits (in 94% of the families in comparison to 38%) [18]. One social worker remarked, “I’m really part of the family–we can talk together and understand together where these problems came from” [18] (p. 6).

The social workers reported changes in their knowledge regarding service users’ life circumstances and changes in how they listen to service users’ stories and interpret them:

One service user was late for a meeting, and the social workers were saying “Excuse me, what is this!? It’s a lack of cooperation!” Now I understand her life circumstances, she has two babies, she needs to come from the other end of the city, there is no direct bus, and so on. The opportunity to stop, look at her circumstances, understand her, that’s the program’s contribution [18] (p. 7).

A recent study conducted among 601 social workers at various SSDs compared those who work in the PAP programs (31% of overall participants) with those who work outside the PAP programs and found 13% higher measures of connectivity with service users among PAP social workers, in terms of regular meetings with families, home visits, and case advocacy practices [22]. Moreover, practitioners in P2 and P3 indicated that they challenged power relations by seeing service users as fully human subjects and by exposing their own vulnerability, as the following quote from one social worker shows: “Service users can see how I feel. I don’t hide it. As I see it, service users are also allowed to be angry … I’m also angry … I’m very authentic. I don’t put on a ‘professional’ façade all the time” [34] (p. 288).

The close experience of standing by service users comes at a cost. Some studies pointed to the emotional and practical burdens connected to the intimate helping relationships that developed with service users whose lives are marked by past and present trauma [18,23,24]. In P2, for example, a social worker explained:

There’s something I have to say about the personal price we pay, a heavy price … When relationships are purely professional, the work is easier. It doesn’t burden your soul, it doesn’t come home with you … As you get closer, people share more things. They share more, we give more. We give more, we have more to do, it’s just a lot more work. Absolutely [35] (p. 291).

Similarly, a social worker in P4 explained:

It’s more difficult to be a social worker in this program than a regular SSD social worker. We have more responsibilities for the families, and the connection is more intensive on a daily basis, on weekends and holidays. You hear all the details of their lives. It’s very hard to cope with this emotionally [24] (p. 26).


Rights-based practice


According to the PAP’s practical model, social rights are mainly addressed via the “active realization of rights” [9] and the provision of material assistance using the family’s “flexible budget” [15,26]. Our sample indicates a thorough adoption of these practices in the PAP programs, e.g., [7,18,23] and beyond the PAP programs in the general operation of some of the SSDs [28,29] thanks to the new special rights social workers’ role, which has been established in approximately half of the SSDs in Israel [16].

The operation of the active realization of rights is evident in all the studies in the sample. For example, in P3, a social worker reported the following: “Many times, I’ve gone with a service user to the bank, talked to the manager … and it’s like [the bank employees] don’t see him, [as if] he is transparent. They don’t look at him or relate to him” [30] (p. 126).

In P2, social workers reported on the subversive practices that they use to meet service users’ needs despite having limited resources:

We go to the State Comptroller Office or the mayor’s Facebook page. Often we more than gently suggest to service users that doing this will help. And then … the mayor contacts us, and I tell him, “I tried to help this service user but I didn’t have enough budget,” and then, as if from heaven, more money arrives, and things come together [36] (p. 8).

Finally, rights realization social workers in P3 addressed the active realization of rights as an integral part of the therapeutic process, integrated with its emotional meanings: “When you help the family members to realize some rights and take care of something they had to take care of … it decreases the overall pressure, and they are freer to take care of the children” [30] (p. 119).

The significance of the active realization of rights is also evident in service users’ outcomes. An analysis of data on 5700 families who participated in P3 between 2015 and 2018 indicates an average increase of 44% in the families’ income from social benefits between the time they entered the program and after two years of being in the program [26] (p. 30). Two other evaluations of P3, conducted in 2018 [19] (n = 346) and 2022 [20] (n = 2029), found a consistent increase in service users’ knowledge of their social rights and their perceived confidence in realizing these rights.

As for the professional use of the flexible budget, it is important to acknowledge the professional narrative of Israeli casework, according to which the provision of material support is detached from direct social work practice in the SSDs and formally provided by the National Insurance Institute [26]. This narrative builds on a conservative perspective on therapeutic processes, highlighting the centrality of emotional over material assistance [14]. Thus, the PAP intervention tool of the flexible budget challenges the common professional perspective of social workers [15].

Although the budget was originally supposed to be flexible and easy to use, an evaluation of P4 found that the actual execution of the flexible budget was complicated due to bureaucratic regulations [24]. These operational obstacles hindered its preliminary purpose as a rights-based practice. Furthermore, the flexible budget raises professional tensions and presents challenges. Qualitative research with 20 social workers in P4 detailed how they moved between conservative and poverty-aware perspectives in terms of the degree of collaboration with service users in determining the purposes of the budget and in terms of trust in service users, as well as in their perception of emotional and material types of support [15].

Nevertheless, the use of the budget in the PAP programs is high. In their analysis of the use of the flexible budget in P3 in two waves of the program from its initiation in 2015 until 2018 (n = 5700), Gal and his colleagues [26] found that 42% of the budget was spent on household needs (appliances, furniture, housing accessories, and installation), 28% on employment (financing of professional courses, counseling and professional guidance, and acquisition of work tools), 10% on the payment of debts (to credit card companies, banks, etc.), and the remaining 20% on health, education, extracurricular activities, and “other” [26] (p. 33). Another significant finding is that the use of the flexible budget was 11% higher in the second wave of the program than in the first. The researchers explain this increase as a result of the learning process required to incorporate such a new intervention tool.

The qualitative studies highlight the significance of the flexible budget for service users and social workers. A service user in P4 related the following: “When the program started, I suddenly felt like I could breathe. Because we received help here with the kid’s nursery school, with food, in the summer, and suddenly I could buy a washing machine … You start to feel like a human being” [23] (p. 26). A social worker in the same program reported: “Before this program, I was always explaining to the families why I can’t do this and that. Today I can [help], and I tell them, ‘think wild, the sky’s the limit.’ And that’s like all my dreams” [24] (p. 26). Furthermore, the social workers demonstrated the interconnectedness of the material and emotional aspects of assistance, as this quote demonstrates: “I brought her a beautiful new baby carriage, and she goes out like a peacock … And after all that [material] help we could talk about a million other things …” [24] (p. 28).

## 4. Conclusions

This article presents a rapid evidence review of programs operating the Poverty-Aware Paradigm in Israeli SSDs in the last seven years. Based on it, we now present several concluding remarks regarding poverty-informed critical social work practice in public social services. First, the PAP programs seem to have made significant achievements in connecting social workers to families living in poverty, restoring trust, and establishing strong helping relationships. Furthermore, rights-based practices seem to be significant for families in both material and emotional aspects [30], enabling meaningful processes of personal and family development [25]. Nevertheless, after an average of two years of intervention in the PAP programs, while the families’ working situations were better, most families remain in fragile financial circumstances [20] and in need of prolonged intensive support that is less available in standard SSD interventions [23,24]. This picture is consistent with previous studies that highlight the neoliberal activation tendency to push people living in poverty to work without fundamentally changing their economic situation [41]. Moreover, it highlights the meaning of life in poverty and the rigidity of the social structure that incarcerates people in it [42]. A two-year program that targets the family as its main unit for intervention is simply not enough to enable those living in poverty to escape it, achieve social justice, or eliminate poverty as a social problem. In other words, PAP practice cannot eliminate poverty by itself, and therefore needs to state the voices and needs of people living in poverty on a more systemic level. To address poverty on a more fundamental level, systemic changes need to be made at the policy level in terms of governmental and municipal commitments to reducing poverty, addressing social mobility mechanisms, changing the social benefits structure, and developing the welfare system in ways that will enable poverty-aware practice based on the SSDs’ operation [43].

Second, the differences in the PAP’s execution between the first wave of families (2015–2017) and the second wave (2018–2020) regarding issues such as social workers’ attitudes [30], the operation of the flexible budget [26], and working with debt [18] point to the existence of a developmental process. This developmental process may be the result of the relative novelty of the PAP model and is relevant for exploring the mechanisms that support and restrict the integration of this critical model into social services. In this regard, institutional backing and integration of the PAP perspective into organizational procedures can play a crucial role in the genuine implementation of the PAP model. This point is of particular importance considering the PAP’s critical aspect, which challenges common societal power relations inherent in the welfare system [9]. A holistic approach to implementing the PAP in ways that incorporate critical discursive positioning and the deconstruction of societal power relations at all levels of SSD operation are key components that can support the PAP’s operation [14,29]. On the other hand, partial implementation of the PAP model, such as the adoption of specific intervention tools or the training of isolated professionals, will probably have less impact on the operation of social services and people living in poverty [30]. Our review demonstrates how limiting the time in the PAP programs and restricting the flexibility of the flexible budget also hinders the operation and outcomes of the PAP model [23,24].

Third, this review allows us to problematize the construction of “success” as it is defined in the PAP programs’ design and evaluation studies. One point is a narrow picture of success, such as focusing on work-market participation while the overall risk of life in poverty prevails, as mentioned earlier. A second point is the assumed automatic correlation between intervention success and decreased investment in families. In other words, common perceptions of success are constructed in terms of service users’ independence, manifested in their ability to cope without professional assistance. This basic perception is evident in the construction of the evaluation studies mainly around micro-level practices and outcomes, such as financial circumstances, and the lack of evaluation that focus on macro-level practice and its outcomes, such as the construction of social mobility mechanisms or the social benefits system. While the normative positioning of independence and personal agency is widely accepted [44], for people living in poverty, the measure of minimal contact with social services may represent their exclusion, mistrust of professional assistance, or fear of paternalistic and intrusive interventions [45]. We would argue that poverty-aware success will consist of standing by people living in poverty in ways that increase interventional measures and maintain high assistance levels for as long as needed, while simultaneously investing in macro-level practices for a more fundamental change [9,43].

## 5. Limitations and Future Research

This review has a few limitations. First, the second-wave interventions of the PAP programs in this review reached their completion during the COVID-19 pandemic, which had substantial effects on social services, social workers, service users, and society at large. As a result, the evaluation studies that addressed the operation and outcomes of these programs during this period were likely affected by the pandemic, and thus their findings should be examined with this point in mind. Second, 15 of the 25 studies in the sample used for this review were written by this article’s authors. Moreover, all three authors have been highly involved in the development, conceptualization, and research of the PAP. This involvement could bias our reading and interpretations of the reviews. Nonetheless, all eight evaluation studies initiated by the Ministry of Welfare [18,19,20,21,22,23,24,25] were external and conducted by independent researchers. Although these external studies were funded by the Welfare Ministry as part of the pilot phase of the programs, they were operated by well-known researchers and agencies, firmly bracketed from the ministry itself. Thus, in this review, we attempted to base our findings on these studies, using our own studies to support and enrich the results of these external evaluations. Furthermore, most of our own studies were conducted with other researchers, who assisted us in bracketing our involvement with the PAP. Third, this review combines quantitative and qualitative research, with more qualitative studies than quantitative ones, complicating the review’s comparability. This sample represents the character of the evaluation and research on social work psychosocial intervention conducted in Israel. To deal with this combination of research methods, we used the qualitative studies to enrich the picture presented by the quantitative ones. Fourth, although the PAP aims to address poverty in both macro and micro practices, this review highlights the micro practice in the PAP programs. This may attest to the direct outcomes which are the focus of the evaluation studies. Furthermore, it can be argued that the adoption and implementation of the PAP by the Welfare Ministry in Israel represents a macro change in working with people living in poverty [6,7]. We believe that this review presents viable findings regarding the operation of the PAP in social services in Israel and provides rich insights into the meanings of critical social work practice and the opportunities it offers. To proceed with these findings, future research should focus on identifying mechanisms or procedures needed to maintain the PAP model’s sustainability over time.

## Figures and Tables

**Figure 1 ijerph-20-00889-f001:**
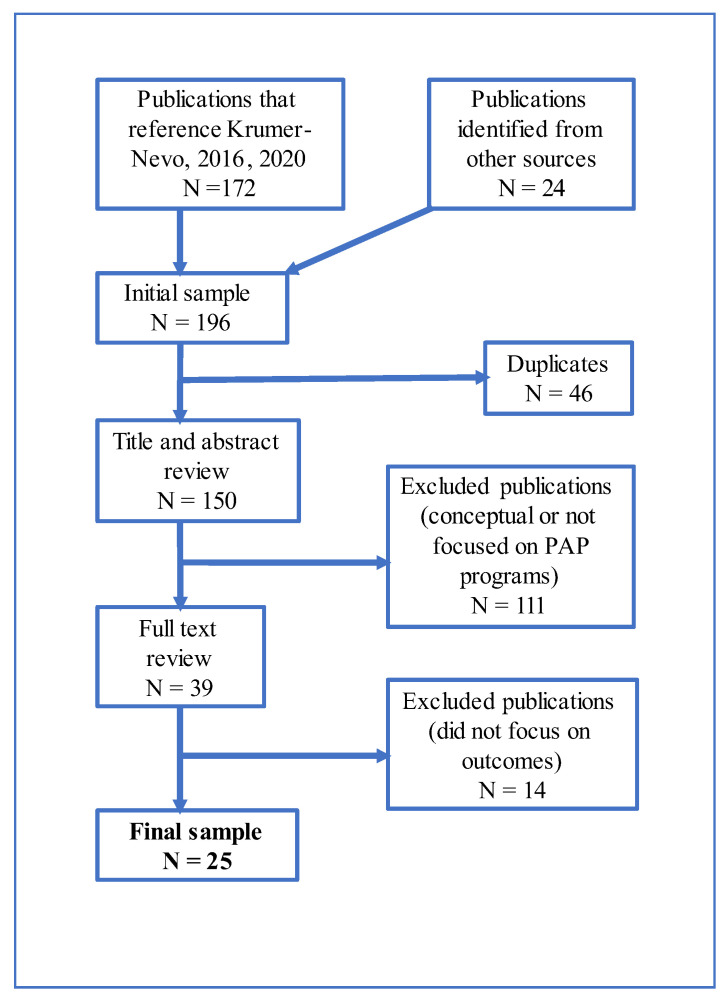
Screening Figure [3,9].

**Table 1 ijerph-20-00889-t001:** Overview of Programs.

Code	Program	Organizational Context	Target Population	No. of Local Authorities in Which the Program Operated	No. of Families that Participated	No. of Social Workers Employed
P1	Casework for Social Change	Fieldwork training program for BA students	Families living in persistent, extreme poverty and branded as “uncooperative” in past attempts to reach out to them in more traditional practice	1	250	72 (students)
P2	Mapa	Ministry of Welfare and Social Affairs (MWSA)–Family services	Families living in persistent, extreme poverty and branded as “uncooperative” in past attempts to reach out to them in more traditional practice	10	150	50
P3	Families First	MWSA–Family services	Families living in poverty with parents who are not coping with mental illness, delinquency, or active addiction	113	14,000	770
P4	Families on the Path to Growth	MWSA–Child protection services	Families with children aged 0–18 identified as being at high risk of child maltreatment—on the verge of removal or reunification	17	330	
P5	Mifgash	MWSA–Child protection services	Families in which children are identified as suffering from neglect	12	303	

**Table 2 ijerph-20-00889-t002:** Sample Overview.

Study	No. in Ref. List	Publication Outlet	Language	Research Site	Method	Data Collection Points	Sample	Study Focus	Findings’ Domain
Benish and Weiss-Gal, 2022	[30]	*Social Security* peer-reviewed journal	Hebrew	P3	Qualitative: semi-structured interviews	1 T	40 frontline social workers	Active actualization role	Impact on SW
2. Gal et al., 2019	[26]	Taub Center for Social Policy Studies in Israel	English	P3	Quantitative: analysis of administrative data	2 Ts	5700 service users	Material assistance	Impact on SW+ SU outcomes
3. Ben-Rabi, 2019	[18]	Myers JDC Brookdale center for applied social research	Hebrew	P2	Mixed methods Quantitative method: questionnaire Qualitative method: focus group interviews	2 Ts	Quantitative sample: 66 service users Qualitative sample: 5 service users and 33 practitioners	Program evaluation	Impact on SW + SU outcomes
4. Weiss-Dagan and Krumer-Nevo, 2022	[31]	*British Journal of Social Work* peer-reviewed journal	English	PAP training courses	Quantitative method: questionnaires	2 Ts	Sample: 92 social workers and 34 social work students	Training	Impact on SW
5. Brand-Levi et al., 2021	[32]	*Health and Social Care in the Community* peer-reviewed journal	English	P2 and P3	Mixed methods Quantitative methods: questionnaire Qualitative methods: structured interviews	1 T	235 service users	Program evaluation	Impact on SW + SU outcomes
6. Brand-Levi et al., 2022	[28]	*British Journal of Social Work* peer-reviewed journal	English	P3	Quantitative methods: questionnaire	2 Ts	159 service users	Program evaluation	SU outcomes
7. Khoury and Krumer-Nevo, 2022	[33]	*International Journal of Social Work* peer-reviewed journal	English	P3	Qualitative method: analysis of supervision protocols		14 social workers	Poverty-aware social work in Arab-Palestinian society	Impact on SW
8. Timor-Shlevin, 2019	[4]	*Social Security* peer-reviewed journal	Hebrew	P2 and P3	Qualitative methods: semi-structured interviews	1 T	41 social work practitioners and managers	Policy implementation	Impact on SW
9. Timor-Shlevin, 2021a	[34]	*British Journal of Social Work* peer-reviewed journal	English	P2 and P3	Qualitative methods: semi-structured interviews	1 T	25 social workers	Poverty-aware practice	Impact on SW
10. Timor-Shlevin, 2021b	[7]	*Social Policy and Administration* peer-reviewed journal	English	P2 and P3	Qualitative methods: semi-structured interviews	1 T	16 senior managers and policymakers	Policy implementation	Impact on SW
11. Timor-Shlevin and Benjamin, 2021	[35]	*Journal of Social Work* peer-reviewed journal	English	P2 and P3	Qualitative methods: semi-structured interviews	1 T	14 social workers	Poverty-aware practice	Impact on SW
12. Timor-Shlevin and Benjamin, 2022	[27]	*European Journal of Social Work* peer-reviewed journal	English	P2 and P3	Qualitative methods: semi-structured interviews	1 T	16 senior managers and policymakers	Policy implementation	Impact on SW
13. Timor-Shlevin, 2022	[36]	*European Journal of Social Work* peer-reviewed journal	English	P2 and P3	Qualitative methods: semi-structured interviews	1 T	25 social work practitioners and managers	Poverty-aware practice	Impact on SW
14. Levy and Freiberg, 2022	[37]	*British Journal of Social Work* peer-reviewed journal	English	P3	Qualitative method: semi-structured interviews	1 T	13 service users	Community practice	Impact on SW + SU outcomes
15. Sabo-Lael et al., 2020	[23]	Myers JDC Brookdale center for applied social research	Hebrew	P4	Mixed methods Quantitative methods–randomized controlled tests and questionnaires Qualitative methods–semi-structured interviews and case studies	2 Ts	Quantitative sample: 224 service users and 100 social workers Qualitative sample: 8 service users and 17 social workers	Program evaluation	Impact on SW + SU outcomes
16. Sorek et al., 2021	[24]	Myers JDC Brookdale center for applied social research	Hebrew	P4	Mixed methods. Quantitative methods: administrative data analysis Qualitative methods: case studies and focus groups	2 Ts	Quantitative sample: 44 service users Qualitative sample: 8 service users and 24 social workers	Program evaluation	Impact on SW + SU outcomes
17. Turjeman and Reuven, 2018	[25]	Western Galilee College academic institution	Hebrew	P5	Mixed methods Quantitative methods: questionnaires Qualitative methods: semi-structured interviews and focus groups	4 Ts	Quantitative sample: 140 practitioners and 175 service users. Qualitative sample: 22 service users and 73 practitioners and policymakers	Program evaluation	Impact on SW + SU outcomes
18. Dank, 2018	[19]	Effective Research for Impact research institute	Hebrew	P3	Mixed methods. Quantitative methods:questionnaires and administrative data Qualitative methods: semi-structured interviews	2 Ts	Quantitative sample: 346 families and 396 social workers Qualitative sample: 19 service users and 99 practitioners and policymakers from senior management	Program evaluation	Impact on SW + SU outcomes
19. Dank, 2022	[20]	Effective Research for Impact research institute	Hebrew	P3	Quantitative methods: questionnaires	2 Ts	2029 families	Program evaluation	SU outcomes
20. Margolin, 2022	[22]	Rashi Foundation research and development	Hebrew	P2, P3, P4, P5	Mixed methods. Quantitative methods: questionnaires Qualitative methods: semi-structured interviews and focus groups	1 T	Quantitative sample: 601 professionals and 1044 service users Qualitative sample: 14 semi-structured interviews and 3 focus groups	Programs evaluations	Impact on SW + SU outcomes
21. Saar-Heiman et al., 2017	[38]	*Child and Family Social Work* peer-reviewed journal	English	P1	Qualitative methods: semi-structured interviews	3 Ts	9 service users	Program evaluation	Impact on SW + SU outcomes
22. Saar-Heiman et al., 2018	[5]	*British Journal of Social Work* peer-reviewed journal	English	P1	Qualitative methods: semi-structured interviews	3 Ts	9 service users	Working in a real-life context	Impact on SW + SU outcomes
23. Saar-Heiman et al., 2022	[29]	*Journal of Social Work* peer-reviewed journal	English	P2 and P3	Qualitative methods: ethnographic observations and participatory workshops	3 Ts	25 social workers and 10 service users	PAP organizational practices	Impact on SW + SU outcomes
24. Saar-Heiman and Krumer-Nevo, 2021	[15]	*American Journal of Psychotherapy* peer-reviewed journal	English	P4	Qualitative methods: semi-structured interviews	1 T	20 interviews with SW	Material assistance	Impact on SW
25. Elisar et al., 2021	[21]	Effective Research for Impact research institute	Hebrew	P3	Mixed methods. Quantitative methods: questionnaires Qualitative methods: semi-structured interviews	3Ts	Quantitative sample: 290 service users Qualitative sample: 11 semi-structured interviews with families	Program evaluation	SU outcomes
**Total**		**Publication**	**Language**	**Research site**	**Method**		**Sample**	**Study focus**	**Finding’s domain**
15 publications written by authors, among them–14 peer-reviewed, 1 research report		* 16 peer-reviewed journals * 9 research reports	* English: 15 * Hebrew: 10	* P1: 2 publications * P2: 11 publications * P3: 17 * P4: 4 * P5: 2 * Training programs:1	* Mixed methods: 8 * Quantitative methods: 3 * Qualitative methods: 14	* 1 T: 11 * 2 Ts: 10 * 3 Ts: 3 *4 Ts: 1	* Quantitative sample: 4612 service users and 1363 social workers * Qualitative sample: 420 service users and 424 social workers and policymakers * Administrative data regarding 5700 service users These numbers are approximate (some studies may overlap)	* Evaluation–10 Direct practice and implementation–8 policy and organizational implementation–4 Training–1 Arab society–1	Impact on SW–10 Impact on SU outcomes–3 Impact on SW and SU outcomes–12

## Data Availability

The data presented in this study is in part openly available in peer-reviewed journals, concerning the cited peer-reviewed articles. Other cited resources, such as research reports and grey literature, which are not openly available, are available on request from the corresponding author.

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
