# Peer review of "Poverty-Aware Programs in Social Service Departments in Israel: A Rapid Evidence Review of Outcomes for Service Users and Social Work Practice"

_ijerph, 2023, doi:10.3390/ijerph20010889_

Round 1

Reviewer 1 Report

This is a fine paper that summarizes well the findings of the implementation and impact of an innovative approach to addressing poverty in social work and social services.  It will certainly contribute much to scholars and practitioners seeking to address this social problem.  The review enables us to move beyond the theory and examine actual results.  It offers a well-thought and strongly-based review of the data and thoughtful conclusions.

I think that this paper can be accepted as is (a recommendation I hardly even make!) but there are a few minor issues that the authors can address: 

1. Define the acronyms in Table 1.

2. The Taub Center description is not correct - see the Center's website.

3. The discussion of the importance of added resources to the program (in comparison to other programs) is very limited (p. 11).  I would add a bit more to this crucial point.

4. I think that it should be underscored somewhere in the paper that only one study quantifies the increase in incomes of participants of the program.

5. What is the Department of Public Inquires?

6. What does the term "conservative" mean in the context of the text on p. 16.

7. Perhaps discuss in the limitations or in the methodology the combination of quantitative and qualitative studies in the review and the relatively limited number of quantitative evaluations.

Author Response

Revisions Letter

We would like to thank the reviewers for their insightful reading of our article and for their helpful comments. This letter lists the revisions made to address the reviewers’ suggestions and comments. It incorporates the reviewers’ comments and is arranged so that after each comment the detailed revision is explained.

Reviewer 1

This is a fine paper that summarizes well the findings of the implementation and impact of an innovative approach to addressing poverty in social work and social services. It will certainly contribute much to scholars and practitioners seeking to address this social problem. The review enables us to move beyond the theory and examine actual results. It offers a well-thought and strongly-based review of the data and thoughtful conclusions.

I think that this paper can be accepted as is (a recommendation I hardly even make!) but there are a few minor issues that the authors can address:

  1. Define the acronyms in Table 1.

# MWSA is the Ministry of Welfare and Social Affairs (The former acronym was incorrect). This explanation was added to the revised version (p. 3).

  1. The Taub Center description is not correct – see the Center's website.

# We amended the description of the Taub Center (p. 6 and 7).

  1. The discussion of the importance of added resources to the program (in comparison to other programs) is very limited (p. 11). I would add a bit more to this crucial point.

# We developed our interpretation of Brand-Levi and her colleague's research (2021) regarding the importance and significance of the PAP's professional position of "standing by" in cultivating a beneficial working alliance with people living in poverty (pp. 11-12).

  1. I think that it should be underscored somewhere in the paper that only one study quantifies the increase in incomes of participants of the program.

# Thanks for that remark that helped us to elucidate this point. It was not only one study that quantified the increase in income; the same increase was found in regard to all the programs by various studies (e.g., Brand-Levi et al., 2022; Elisae et al., 2021; Sorek et al., 2021). We revised this point on pp. 12-13.

  1. What is the Department of Public Inquires?

# It should have been presented as the State Comptroller Office. We corrected this quote (p. 16).

  1. What does the term "conservative" mean in the context of the text on p. 16.

# We added an explanation regarding the conservative perspective of therapeutic processes, according to which emotional assistance is more important than material assistance since it creates a more significant change (p. 16).

  1. Perhaps discuss in the limitations or in the methodology the combination of quantitative and qualitative studies in the review and the relatively limited number of quantitative evaluations.

# We addressed this point in the limitations section as followed: This review combines quantitative and qualitative research, with more qualitative studies than quantitative ones, complicating the review's comparability. This sample represents the character of evaluation and research on social work psychosocial intervention done in Israel. To deal with this combination of research methods, we used the qualitative studies to enrich the picture presented by the quantitative ones (p. 18).

Reviewer 2 Report

Thank you for the opportunity to review this manuscript. The implementation of the Poverty-Aware Paradigm (PAP) is indeed one of very few examples in which public welfare services adopted critically informed programs, and possibly the most established such example to address poverty in recent years. Therefor a review of the evidence is both relevant and timely. Evidence from 25 studies and evaluations of programs that implemented PAP principals in social service departments in Israel is presented with a wide scope of both qualitative and quantitative data. The manuscript analyses this rich data, it is informative, well written, and concise. I have a few questions and clarifications for authors to consider:

The title refers to implementation and section 1.2 briefly describes this. However, the 2 research questions focus on the outcomes. Following this, the findings also report the PAP program outcomes for service users and impact on social workers attitudes (which might also be viewed as an outcome of the training component), but not the implementation. Implementation of practice is mentioned again on section 3.2.2. referring to social workers attitudes. The inconsistent framing is confusing.

The PAP is presented as unique in that it established strategies of macro-level practice. However, this rapid review is focused on individual and family-based practice.

The findings section presents rich data that touches on multiple aspects of the PAP, offering concrete examples that address the program's strengths, challenges, and consequences. The mixed findings regarding services users' financial circumstances are interesting from a structural perspective but go somewhat overlooked both in the findings and in the conclusion section. While it is promising that many users entered the labor market following the implementation of the PAP, the outcomes regarding their financial situation offer little encouragement [20]. This is consistent with many previous studies indicating how welfare programs in neoliberal economies tend to push clients to work although often keeping them poor and struggling.

The conclusions critically deal with the findings. While the third point in this section is insightful and valuable, a better connection to the findings can strengthen this claim.

Methods - Rapid reviews are not commonly executed in SW, however this choice is explained convincingly. 1) Figure 1 illustrates the screening process. There are 111 excluded publications after title and abstract review. I could not find an explanation in the figure or a textual description of the process by which they were excluded. 2) Table 2 clearly details the sample overview. It would be clearer if studies were linked with the references section (especially for the many studies authored by any of the current manuscript authors). As some of the program evaluations appear to have not been published in academic outlets, linking these reports would also add to the study rigor and promote transparency.  3) It is not clear which research report "initiated and published by a research center" noted as [26] is mentioned in lines 175-177 since there are 25 studies. If this is not a typo, an explanation is needed. 4) There is no report of how many reviewers (authors or others) preformed the screening, were they independently reviewed? How were differences (if any) reconciled. Adding this information may enhance rigor and perhaps resolve some of the study limitations detailed later.

Finally, the study limitations clearly report the authors deep involvement with the development, implementation and research of the programs and consider the biases. Regarding the evaluation initiated by the Israeli Ministry of Welfare – to the extent that any of them were commissioned as part of the program this should also be acknowledged in the limitations and possibly also in the funding section.

Author Response

Revisions Letter

We would like to thank the reviewers for their insightful reading of our article and for their helpful comments. This letter lists the revisions made to address the reviewers’ suggestions and comments. It incorporates the reviewers’ comments and is arranged so that after each comment the detailed revision is explained.

Reviewer 2

Thank you for the opportunity to review this manuscript. The implementation of the Poverty-Aware Paradigm (PAP) is indeed one of very few examples in which public welfare services adopted critically informed programs, and possibly the most established such example to address poverty in recent years. Therefor a review of the evidence is both relevant and timely. Evidence from 25 studies and evaluations of programs that implemented PAP principals in social service departments in Israel is presented with a wide scope of both qualitative and quantitative data. The manuscript analyses this rich data, it is informative, well written, and concise. I have a few questions and clarifications for authors to consider:

  1. The title refers to implementation and section 1.2 briefly describes this. However, the 2 research questions focus on the outcomes. Following this, the findings also report the PAP program outcomes for service users and impact on social workers attitudes (which might also be viewed as an outcome of the training component), but not the implementation. Implementation of practice is mentioned again on section 3.2.2. referring to social workers attitudes. The inconsistent framing is confusing.

# Your remark allowed us to be more precise regarding our contribution. As we wanted to detail the impact and outcomes of the actual operation of the PAP programs, we revised the article's aim and headline, dismissing the term "implementation", and focusing on outcomes. The revised headline of the article is: "Poverty-Aware Programs in Social Service Departments in Israel: A Rapid Evidence Review of Outcomes for Service Users and Social Work Practice".

  1. The PAP is presented as unique in that it established strategies of macro-level practice. However, this rapid review is focused on individual and family-based practice.

# Thanks for noticing this point. It is true that PAP aims to link micro and macro level practice. As far as we know the field, there are indeed various initiatives of macro-practice in the PAP programs. However, macro-practice is not included in the evaluation studies that we reviewed, that are interested mainly in outcomes in the lives of service users and in the professional practice. We might consider the change in the profession (the development of a new role of rights social workers, the addition of budget for families, and the development of a training and supervision in the SSDs) as an example of macro-practice that is supposed to help social workers to stand by service users. As we wrote, other macro-practice initiatives, that dealt directly with poverty, were not included in the evaluation studies and deserve their own methodical scrutiny.    We addressed the lack of attention to macro practices in the evaluation studies we reviewed in the conclusion and limitation sections.

  1. The findings section presents rich data that touches on multiple aspects of the PAP, offering concrete examples that address the program's strengths, challenges, and consequences. The mixed findings regarding services users' financial circumstances are interesting from a structural perspective but go somewhat overlooked both in the findings and in the conclusion section. While it is promising that many users entered the labor market following the implementation of the PAP, the outcomes regarding their financial situation offer little encouragement [20]. This is consistent with many previous studies indicating how welfare programs in neoliberal economies tend to push clients to work although often keeping them poor and struggling.

# Thanks for that remark! We addressed this point in the conclusion section, highlighting the need to address poverty not only at the micro level by pushing people to the working market, through several changes at the macro level.

  1. The conclusions critically deal with the findings. While the third point in this section is insightful and valuable, a better connection to the findings can strengthen this claim.

# We interpreted the meaning of success as service users coping without assistance from the construction of the evaluation mainly around micro-level practices and outcomes, such as financial circumstances, and the lack of macro-level outcomes, such as the construction of social mobility mechanisms or the social benefits system. This explanation is added to the conclusion section (p. 18).

  1. Methods – Rapid reviews are not commonly executed in SW, however this choice is explained convincingly. 1) Figure 1 illustrates the screening process. There are 111 excluded publications after title and abstract review. I could not find an explanation in the figure or a textual description of the process by which they were excluded.

# Thanks for noticing and pointing this out. We added explanations regarding the screening procedure and the excluded publications, (because they were conceptual or did not focus on PAP programs and settings), both in the text (p. 4) and in the screening figure (p. 5).

  1. 2) Table 2 clearly details the sample overview. It would be clearer if studies were linked with the references section (especially for the many studies authored by any of the current manuscript authors). As some of the program evaluations appear to have not been published in academic outlets, linking these reports would also add to the study rigor and promote transparency.

# We added the publications numbers in the reference list to Table 2.

  1. 3) It is not clear which research report "initiated and published by a research center" noted as [26] is mentioned in lines 175-177 since there are 25 studies. If this is not a typo, an explanation is needed.

# There are 25 publications in the review; however, the number 26 here represents the serial number in the reference list, which naturally contains more references than the reviewed publications. This specific publication is Gal et al. (2019), listed as no. 2 in Table 2, and no. 26 in the reference list.

  1. 4) There is no report of how many reviewers (authors or others) preformed the screening, were they independently reviewed? How were differences (if any) reconciled. Adding this information may enhance rigor and perhaps resolve some of the study limitations detailed later.

# A more detailed explanation of the author's part in the screening process was added to the method section (p. 4).

  1. Finally, the study limitations clearly report the authors deep involvement with the development, implementation and research of the programs and consider the biases. Regarding the evaluation initiated by the Israeli Ministry of Welfare – to the extent that any of them were commissioned as part of the program this should also be acknowledged in the limitations and possibly also in the funding section.

# We acknowledged this point in the limitation section (p. 19).